# Assessment of Back-Squat Performance at Submaximal Loads: Is the Reliability Affected by the Variable, Exercise Technique, or Repetition Criterion?

**DOI:** 10.3390/ijerph18094626

**Published:** 2021-04-27

**Authors:** Alejandro Pérez-Castilla, Danica Janicijevic, Zeki Akyildiz, Deniz Senturk, Amador García-Ramos

**Affiliations:** 1Department of Physical Education and Sport, Faculty of Sport Sciences, University of Granada, 18010 Granada, Spain; amagr@ugr.es; 2Research Academy of Human Biomechanics, The Affiliated Hospital of Medical School of Ningbo University, Ningbo University, Ningbo 315020, China; jan.danica@gmail.com; 3Faculty of Sports Science, Ningbo University, Ningbo 315211, China; 4Faculty of Sport and Physical Education, The Research Centre, University of Belgrade, 11000 Belgrade, Serbia; 5Movement and Training Science Department, Gazi University, Ankara 06560, Turkey; zekiakyldz@hotmail.com; 6School of physical education of sports department, Gelişim University, Istanbul 34310, Turkey; dsenturk@gelisim.edu.tr; 7Faculty of Education, Department of Sports Sciences and Physical Conditioning, Universidad Católica de la Santísima Concepción, Concepción 4070129, Chile

**Keywords:** linear position transducer, power, resistance training, stretch-shortening cycle, velocity-based training

## Abstract

This study aimed to compare the between-session reliability of different performance variables during 2 variants of the Smith machine back-squat exercise. Twenty-six male wrestlers performed 5 testing sessions (a 1-repetition maximum [1RM] session, and 4 experimental sessions [2 with the pause and 2 with the rebound technique]). Each experimental session consisted of performing 3 repetitions against 5 loads (45–55–65–75–85% of the 1RM). Mean velocity (MV), mean power (MP), peak velocity (PV), and peak power (PP) variables were recorded by a linear position transducer (GymAware PowerTool). The best and average scores of the 3 repetitions were considered for statistical analyses. The coefficient of variation (CV) ranged from 3.89% (best PV score at 55% 1 RM using the pause technique) to 10.29% (average PP score at 85% 1 RM using the rebound technique). PP showed a lower reliability than MV, MP, and PV (CV_ratio_ ≥ 1.26). The reliability was comparable between the exercise techniques (CV_ratio_ = 1.08) and between the best and average scores (CV_ratio_ = 1.04). These results discourage the use of PP to assess back-squat performance at submaximal loads. The remaining variables (MV, MP, or PV), exercise techniques (pause or rebound), and repetition criteria (best score or average score) can be indistinctly used due to their acceptable and comparable reliability.

## 1. Introduction

One basic consideration when designing any resistance training program is to establish appropriate testing procedures to monitor physical performance throughout the training cycle [1]. Nowadays, practitioners can accurately monitor different performance variables (i.e., force, velocity, and power) during many resistance training exercises due to the advancement and proliferation of sports technology [2,3,4]. Linear position transducers (LPTs) are undoubtedly the devices that have attracted the most interest in the strength and conditioning community [4]. Mean and peak values of velocity and power are commonly collected with LPTs against the same absolute loads before and after training to detect changes in strength performance [5,6,7,8]. Previous studies have reported an acceptable reliability for velocity and power variables collected across a wide range of submaximal loads in different exercises such as the bench press [9,10,11], back-squat [10,12,13], and deadlift [12,14,15]. However, there is no definitive evidence regarding which of the different velocity and power variables (mean velocity [MV] vs. mean power [MP] vs. peak velocity [PV] vs. peak power [PP]) that can be collected with a LPT provides the most reproducible assessment of performance in basic resistance training exercises [9,10,12,14].

The back-squat is one of the most effective exercises to strengthen the lower limbs and prevent injuries [16]. Additionally, lower-body strength gains following back-squat training have been shown to positively transfer to athletic performance during short duration high-intensity actions such as jumping and sprinting [17,18]. The back-squat exercise can be performed using the pause (i.e., a pause of 1–4 s is implemented between the lowering and lifting phases) or rebound techniques (i.e., the lifting phase is performed immediately after the lowering phase using the stretch-shortening cycle) [19,20]. Previous studies have reported acceptable levels of reliability for the measurements of MV, PV, MP, and PP (coefficient of variation [CV] ≤ 8.40%; intraclass correlation coefficient [ICC] ≥ 0.58) collected by the LPT GymAware PowerTool across a wide range of relative loads (20–90% of one-repetition maximum [1RM]) during the free-weight back-squat exercise performed with the rebound technique [10,12]. More importantly, although the reliability was not explicitly compared in these studies, MV (CV range = 3.10–6.70%) and PV (CV range = 2.60–5.50%) were generally collected with a higher reliability than MP (CV range = 3.00–8.00%) and PP (CV range = 3.60–7.40%). Moreover, the back-squat exercise has been shown to report more consistent velocity measurements using the pause technique compared to the rebound technique [19]. It should be also noted that machine-based exercises can provide more reliable measures of velocity than free-weight exercises [21], and, consequently, it is common that the testing procedures of mechanical performance against submaximal loads are performed in a Smith machine to reduce the noise of the test [11,22,23,24]. Therefore, it seems important to compare the reliability of the different performance variables during the back-squat exercise performed in a more controlled environment (i.e., Smith machine) using both the pause and rebound techniques.

Routine testing procedures often require participants to perform multiple repetitions against the same load [10,11,14,22]. Practitioners then need to decide whether the best (i.e., the highest) or the average value of these repetitions will be used for comparative analyses. Some studies have used the average score of all repetitions [10,11,14,15,22], other studies have used the best score [23,25], and others did not directly specify the repetition criteria used for data analysis [12,19]. Given the diversity of repetition criteria, previous research has explored whether the best or average scores are more appropriate as an indicator of physical performance [26,27,28,29]. For example, Claudino et al. [27] showed that the average countermovement jump (CMJ) height was more sensitive than the highest CMJ height for monitoring the neuromuscular status. Similarly, Bishop et al. [26] reported greater consistency for the inter-limb differences evaluated in the isometric squat, CMJ, and drop jump exercises using the average score compared to the best score of each limb. However, Rios et al. [28] found that the reliability of throwing velocity in different handball throwing tests was comparable when the best score of 4 trials, average score of 4 trials, and the average score of the 3 best trials were considered for the analyses. Therefore, to shed more light on this topic, it seems important to elucidate whether the reliability of the different performance variables collected during the back-squat exercise are affected by the repetition criterion.

To address these research gaps, mean and peak values of velocity and power were collected in the present study on separate occasions across a range of submaximal loads (45–55–65–75–85% 1RM) during the Smith machine back-squat exercise performed using the pause and rebound techniques. Specifically, this study aimed to compare the between-session reliability between (i) 4 performance variables (MV vs. MP vs. PV vs. PP), (ii) 2 exercise techniques (pause vs. rebound), and (iii) 2 repetition criteria (best vs. average scores). It was hypothesized that (i) MV and PV would be the most reliable variables, followed by MP, and finally PP [10,12], (ii) the pause technique would provide more reliable outcomes than the rebound technique [19], and (iii) the average score would provide comparable or greater reliability outcomes than the best score [26,28].

## 2. Materials and Methods

### 2.1. Participants

Twenty-six male wrestlers volunteered to participate in this study (age = 19.1 ± 3.3 years [range = 16–32 years], stature = 1.71 ± 0.12 m; body mass = 74.7 ± 11.8 kg; back-squat 1RM = 149.8 ± 23.2 kg). All participants reported using the back-squat exercise in their training programs (4.8 ± 2.4 years) and having competed in national competitions. Participants were not allowed to perform any strenuous physical activity during the 24 h preceding each testing session. All participants were free from health problems or musculoskeletal injuries at the time of data acquisition. Before testing, participants were informed about the research purpose and procedures, and they or their legal guardians (for participants younger than 18 years) gave written consent to participate in the study. The study protocol adhered to the tenets of the Declaration of Helsinki and was approved by the Institutional Review Board (IRB approval: 687/CEIH/2018).

### 2.2. Study Design

A repeated-measures design was used to compare the between-session reliability of different performance variables during 2 variants of the Smith machine back-squat exercise. After a 1RM testing session, participants undertook 4 experimental sessions (twice per week) over 2 consecutive weeks. In a counterbalanced order, participants performed the back-squat exercise using the pause or rebound techniques. The 2 sessions of the same technique were performed in the same week separated by at least 48 h of rest. All testing sessions were performed in a Smith machine (Technogym, Gambettola, Italy) and were conducted at the same time of day for each participant (±1 h) and under similar environmental conditions (~22 °C and ~60% humidity).

### 2.3. Procedures

A preliminary session was used to familiarize the participants with the lifting of submaximal loads at maximal velocity during the pause and rebound back-squat exercises, and to determine the back-squat 1 RM using the pause technique. Stature and body mass (Seca model 654, Seca^®^, Hamburg, Germany) were measured at the beginning of the first session. The warm-up consisted of running on a treadmill for 10 min at 6.5 km·h^−1^, dynamic stretching, and 1 set of 5 repetitions with an external load of 17 kg (mass of the unloaded Smith machine barbell). Briefly, the 1 RM testing protocol consisted of performing 3–5 repetitions at ~50–80% of participants’ self-perceived 1 RM, followed by 2–5 single attempts to determine the 1RM strength. Three minutes of rest was given between sets and 1 RM attempts.

The main experimental sessions began with the same warm-up described for the preliminary session. After warming-up, participants rested for 5 min and then performed the back-squat exercise using either the pause or rebound techniques against 5 loads (45–55–65–75–85% 1RM). The 5 loads were applied in an incremental order and 3 repetitions were executed with each load. Inter-repetition rest was set to 10 s and inter-set rest was fixed to 3 min. Participants received MV feedback immediately after performing each repetition and were encouraged to perform all repetitions at maximal intended velocity [30]. Two spotters were standing on each side of the barbell to ensure safety. The specific characteristics of the 2 back-squat techniques are provided below:

Back-squat with the pause technique. Participants initiated the movement in a fully extended position with the feet shoulder-width apart and the barbell held across the back at the level of the acromion (“high-bar position”). From this position, they were required to descend in a continuous motion before reaching 90° of knee flexion, hold this position for ~2 s, and then return to the initial position as fast as possible. The squat depth was individually controlled using an elastic cord positioned under participants’ hips [31]. Participants were instructed to keep constant downward pressure on the barbell throughout the whole movement, and they were not allowed to jump off the ground.

Back-squat with rebound technique. The execution technique was identical to the back-squat exercise performed with the pause technique, but in this case, participants initiated the lifting phase immediately after reaching the 90° knee flexion.

### 2.4. Data Acquisition

A validated linear position transducer (GymAware PowerTool, Kinetic Performance Technologies, Canberra, Australia) was used to automatically measure the MV, MP, PV, and PP of all repetitions [4]. The cable of the linear position transducer was attached vertically to the right side of the barbell using a velcro strap. The device was sampled with a level-crossing detection method and stamped with a high resolution of 35 microseconds the changes in barbell position, which were down-sampled to 50 Hz for analysis [14,32]. Velocity and acceleration data were calculated from the first and second derivate of the change in barbell position with respect to time, while force data were calculated by multiplying the lifted mass by the total acceleration (gravity + acceleration of the barbell). Finally, power was computed as the product of force and velocity. Data obtained from the device were transmitted via Bluetooth^TM^ to a tablet (iPad, Apple Inc., Cupertino, CA, USA) using the GymAware v2.8 app, and to the online cloud before being exported to Microsoft Excel (Microsoft Corporation, Redmond, WA, USA) and prepared for further analysis. Three repetitions were performed with each load in each session, and both the highest (best) score and the average score of the 3 repetitions were used for statistical analyses.

### 2.5. Statistical Analyses

Descriptive data are presented as means and SDs, whereas the CV and ICC are presented through their median values and range. The normal distribution of the data was confirmed using the Shapiro–Wilk test (*p* > 0.05). Paired samples *t*-tests and standardized mean difference (Cohen’s d effect size [ES]) were used to compare the magnitude of the different variables between both testing sessions. The criteria to interpret the magnitude of the ES was the following: trivial (<0.20), small (0.20–0.59), moderate (0.60–1.19), large (1.20–2.00), or very large (>2.00) [33]. Between-session reliability was assessed by the standard error of measurement (SEM), the CV (standard error of measurement/participants’ mean score × 100) and the ICC (model 3.1) with their corresponding 95% confidence intervals. Acceptable reliability was determined as a CV < 10% and ICC > 0.70 [34]. The ratio between 2 CVs was used to compare the reliability between the 4 variables (MV, MP, PV, and PP), 2 exercise techniques (pause and rebound), and 2 repetition criteria (best and average). The smallest important ratio between 2 CVs was considered to be higher than 1.15 [35]. The reliability analysis was performed through a custom spreadsheet [36]. Alpha was set at 0.05.

## 3. Results

Between-session reliability of MV, MP, PV, and PP variables obtained from the best and average scores during the back-squat exercise performed with the pause or rebound techniques are depicted in Table 1 and Table 2, respectively. No significant differences (*p* > 0.05 in 39 out of 40 comparisons for the pause technique, and 26 out of 40 comparisons for the rebound technique) and *trivial* to *small* differences (ES ≤ 0.48) were generally observed for the different variables between both testing sessions. All variables presented an acceptable absolute reliability for both exercise techniques and repetition criteria (CV = 5.95% [3.89–9.31%]), with the only exception of the average score of PP attained at 85% 1 RM using the rebound technique (CV = 10.29%). Regardless of the exercise technique and repetition criteria, the relative reliability was always acceptable for MP and PP variables (ICC = 0.92 [0.83–0.96]), but generally unacceptable for MV and PV variables (ICC = 0.65 [0.52–0.77]).

Regarding the reliability comparisons, the main findings revealed that the reliability was: (i) comparable between MV, MP, and PV variables (CV_ratio_ ≤ 1.10), but lower for PP compared to MV (CV_ratio_ = 1.37), MP (CV_ratio_ = 1.26), and PV (CV_ratio_ = 1.39) (Figure 1); (ii) comparable between the pause and rebound techniques (CV_ratio_ = 1.08) (the exception was observed for MP in which the pause technique was more reliable than the rebound technique; CV_ratio_ = 1.23) (Figure 2); and (iii) comparable between best and average scores (CV_ratio_ = 1.04) (Figure 3).

## 4. Discussion

This study was designed to compare during the Smith machine back-squat exercise performed across a range of submaximal loads the between-session reliability of different performance variables (MV, MP, PV, and PP), repetition criteria (best and average scores), and exercise techniques (pause and rebound). The main findings revealed that: (i) all performance variables generally presented an acceptable absolute reliability (CV < 10%) for both exercise techniques and repetition criteria, but the reliability of PP was somewhat lower; and (ii) both exercise techniques and repetition criteria provided comparable reliability outcomes. From a reliability standpoint, these results indicate that the PP should be discouraged when assessing back-squat performance at submaximal loads. The remaining variables (MV, MP, or PV), exercise techniques (pause or rebound), and repetition criteria (best score or average score) can be indistinctly used due to their acceptable and comparable reliability.

A basic property of any physical test is the reliability of the measurement [37]. In line with previous studies [10,12,13,19], all performance variables analyzed in the present study revealed an acceptable reliability (CV < 10% and ICC > 0.70) during the back-squat exercise, but the relative reliability (i.e., ICC values) of the velocity variables was generally unacceptable. It is worth noting that ICCs values are sensitive to the heterogeneity of the sample tested (more heterogeneity = higher ICCs value) [37]. Therefore, since all participants lifted the same relative loads (% 1RM), which are supposed to be lifted at similar velocities [13,20,38], the lower heterogeneity of velocity variables (between-participants CV range = 6.1–13.4%) compared to power variables (between-participants CV range = 20.2–24.8%) may explain the differences in the relative reliability. Note that power is computed considering both the lifting velocity and the absolute load lifted, and because the absolute load lifted was different across participants the heterogeneity of power variables was higher. Similar findings (i.e., higher absolute reliability for velocity variables and higher relative reliability for power variables) have been reported when the same relative loads are lifted during the bench press [9] and deadlift [14] exercises. Thompson et al. [38] have also shown that the reliability of velocity variables collected during the back-squat exercise decreased with the increment of the load ranging from acceptable (CV = 8.2% for MV and 9.5% for PV at 30% 1 RM) to unacceptable (CV = 24.2% for MV and 27.8% for PV at 100% 1 RM). Therefore, it is not surprising that in the present study, the absolute reliability for the velocity and power variables was consistent across all relative loads but slightly lower for the heaviest load (85% 1RM). These results collectively suggest that practitioners should be more careful when tracking changes in performance against heavy loads (≥85% 1 RM) during the back-squat exercise due to lower reproducibility of measurement.

Our first hypothesis was only partially supported since the reliability of MV, MP, and PV variables was comparable, but PP showed a slightly lower reliability. These results are in agreement with the findings of Grgic et al. [12], who generally found a comparable reliability between the 4 performance variables (MV, PV, MP, and PP) during the free-weight back-squat exercise performed using the rebound technique (CV_ratio_ ≤ 1.14), except for PV which was more reliable than PP (CV_ratio_ = 1.24). Our results are also partially in line with the findings of Orange et al. [10] who observed a higher reliability during the free-weight back-squat exercise performed using the rebound technique for MV and PV variables compared to PP (CV_ratio_ = 1.24 and 1.30, respectively) and MP (CV_ratio_ = 1.47 and 1.54, respectively). Although the same measurement system (GymAware PowerTool) was used in all abovementioned studies, the slight discrepancies in the results may be attributed to certain methodological factors such as the squat depth (parallel-squat vs. half-squat), the equipment (free-weight vs. Smith machine), or the sample of participants tested because in all studies the biological error was considered. In either case, as has been shown for other exercises [9,14], the available literature suggests that velocity variables can be obtained with a higher reliability than power variables (see Figure 1 for further details). This is likely caused by the higher manipulation of the raw displacement-time data–recorded by linear position transducers–needed to obtain power outputs in comparison to velocity outputs [9]. The lower reliability observed for PP supports the widespread thinking that mean values may represent better non-aerial movements such as back-squat, while peak values may be more relevant for ballistic movements such as squat jumps [10,13,22]. However, as it has been reported in this and previous studies conducted with the back-squat exercise [10,12], the reliability of PV does not differ with respect to MV and MP.

Rejecting our second hypothesis, no meaningful differences in reliability were observed between the pause and rebound techniques. These results are in disagreement with the findings of Pallarés et al. [19], who reported a higher reliability for the full spectrum of velocities during the back-squat exercise performed using the pause technique compared to the rebound technique (CV = 2.9% vs. 3.9%; *p* = 0.010). The discrepancy between the results of the present study and Pallarés’s et al. study [19] is likely caused by the instructions given to the participants regarding the execution of the lowering phase during the rebound technique. Specifically, Pallarés et al. [19] controlled the duration of the lowering phase with a real time auditory feedback, while in the present study participants were instructed to perform the lowering phase at a fast and self-controlled velocity. It is plausible that the reliability could be compromised by redirecting the focus of attention to the auditory signal rather than maximizing performance in the subsequent lifting phase. This assumption is further supported by a recent study conducted in the bench press exercise that reported a lower reliability of velocity variables when the velocity of the lowering phase was externally controlled in comparison to the pause technique or performing the lowering phase at a fast and self-controlled velocity [24]. Based on the prevailing evidence, it is likely that resistance exercises performed with the externally-controlled rebound technique may increase the variability of velocity outputs and reduce its ecological validity since most sports activities are performed with fast stretch-shortening cycles or at least the velocity of the lowering phase is not externally stipulated [24]. Therefore, it seems that the pause technique does not provide more reproducible velocity outputs than the rebound technique when the lowering phase is performed at a fast and self-controlled velocity.

Confirming our third hypothesis, no meaningful differences in reliability were observed between the criteria (best or average score) used for data analysis. These results are in agreement with the study of Rios et al. [28], who did not find meaningful differences in reliability between the repetition criteria used for assessing throwing velocity in different handball throwing tests. However, our results are in disagreement with the findings of Claudino et al. [27] and Bishop et al. [26], who observed a greater sensitivity of the CMJ height to monitor the neuromuscular status or a greater consistency to calculate isometric squat, CMJ, and drop jump asymmetries using the average score instead of the best score, respectively. It is plausible that variables that are obtained with a high reliability are not affected by the repetition criterion; however, the repetition criterion based on average scores could be a more robust indicator of performance for variables that are obtained with a lower reliability since the best scores can be affected by outliers related to erroneous lifts [39]. To avoid this potential problem, which was not observed in the present study, researchers and practitioners must ensure that no more than 10% differences are observed between trials, and if observed participants should perform additional repetitions and discard the extreme values [29].

This study presents several limitations that should be acknowledged. For example, the use of a Smith machine, which restricts the movement of the barbell to the vertical direction, may limit the ecological validity of our findings because athletes typically perform the back-squat exercise with free-weights. However, since it has been shown that machine-based exercises provide more reliable measures of movement velocity than free-weight exercises [21], we decided to use the Smith machine to eliminate the possible confounding factors that could be present during free-weight exercises (e.g., horizontal movements of the barbell). In addition, although it is less frequent in training, resistance training exercises are often performed in a Smith machine for testing purposes [11,22,23,24]. It should be also noted that the application of the loads in an incremental order could have influenced the reliability comparisons across the loads due to a possible effect of potentiation or fatigue. However, it should be noted that the comparison of reliability across the loads was not a specific aim of the present study, while we adopted an incremental order of the loads to be consistent with previous studies that have explored the reliability of different mechanical outputs during the back-squat exercise [10,12].

## 5. Conclusions

From a reliability standpoint, researchers and practitioners are discouraged from using PP when assessing back-squat performance at submaximal loads, while the remaining performance variables (MV, MP, or PV) can be indistinctly used due to their acceptable and comparable reliability. Besides, there are no differences in reliability between the pause and rebound techniques of the back-squat exercise. Therefore, when the lowering phase is performed at a fast and self-controlled velocity, practitioners can expect a comparable reliability for the pause and rebound techniques. Finally, no meaningful differences were observed in reliability between the best and average scores and this is in line with previous studies that have also explored variables that are obtained with a high reliability (CV < 10%). However, the average score could be preferable when the intra-individual variability is higher (CV > 10%) because there are greater chances that the best score is affected by outliers.

## Figures and Tables

**Figure 1 ijerph-18-04626-f001:**
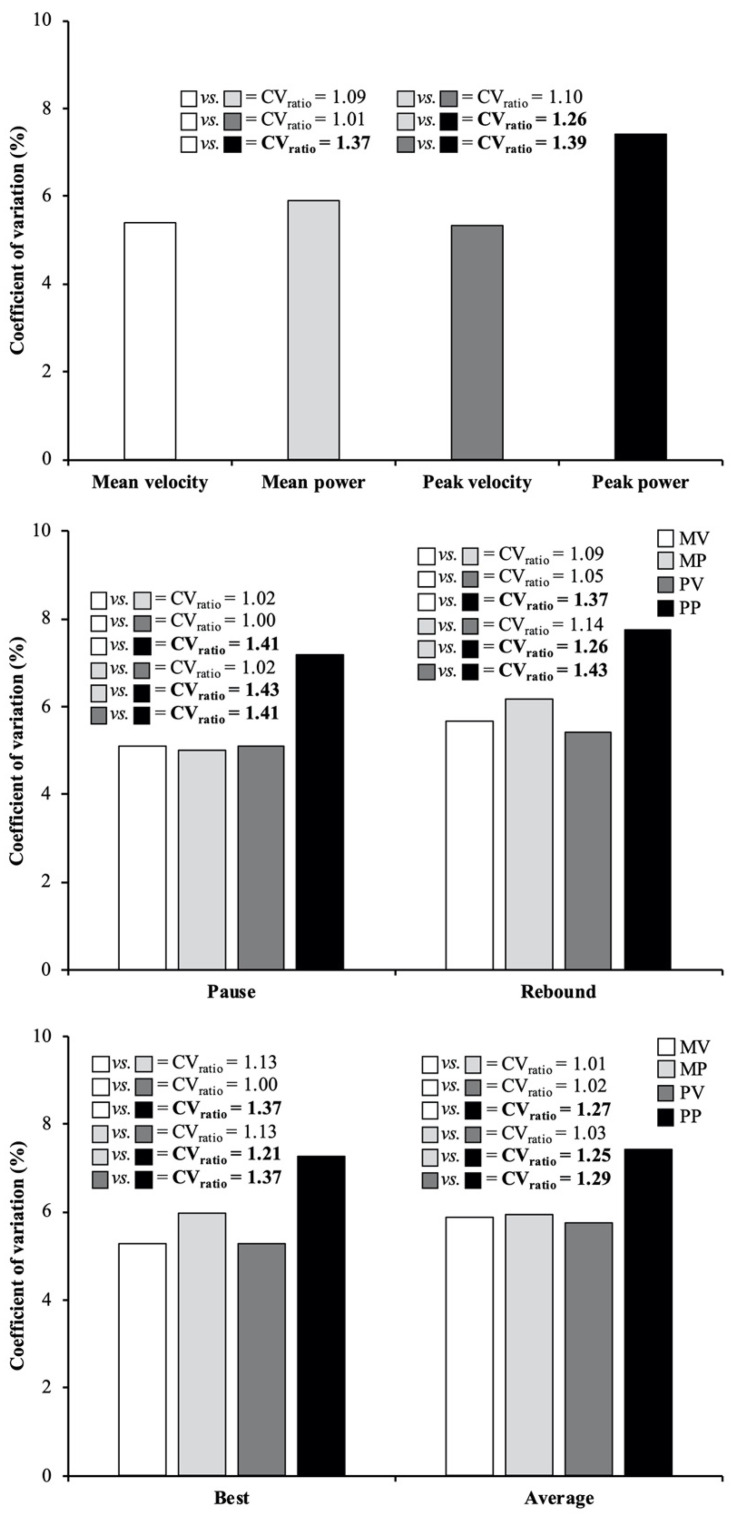
Reliability comparisons between mean velocity (MV; white bars), mean power (MP; light gray bars), peak velocity (PV; dark gray bars), and peak power (PP; black bars) variables collected during the back-squat exercise. Bars represent the median coefficient of variation value obtained combining the 2 exercise techniques and 2 repetition criteria (upper panel), the 2 repetition criteria separately for each exercise technique (middle panel), and the 2 exercise techniques separately for each repetition criterion (lower panel). Numbers depict the ratio between 2 coefficients of variation (CV_ratio_ = higher/lower value), while meaningful differences in reliability are indicated in bold (CV_ratio_ > 1.15).

**Figure 2 ijerph-18-04626-f002:**
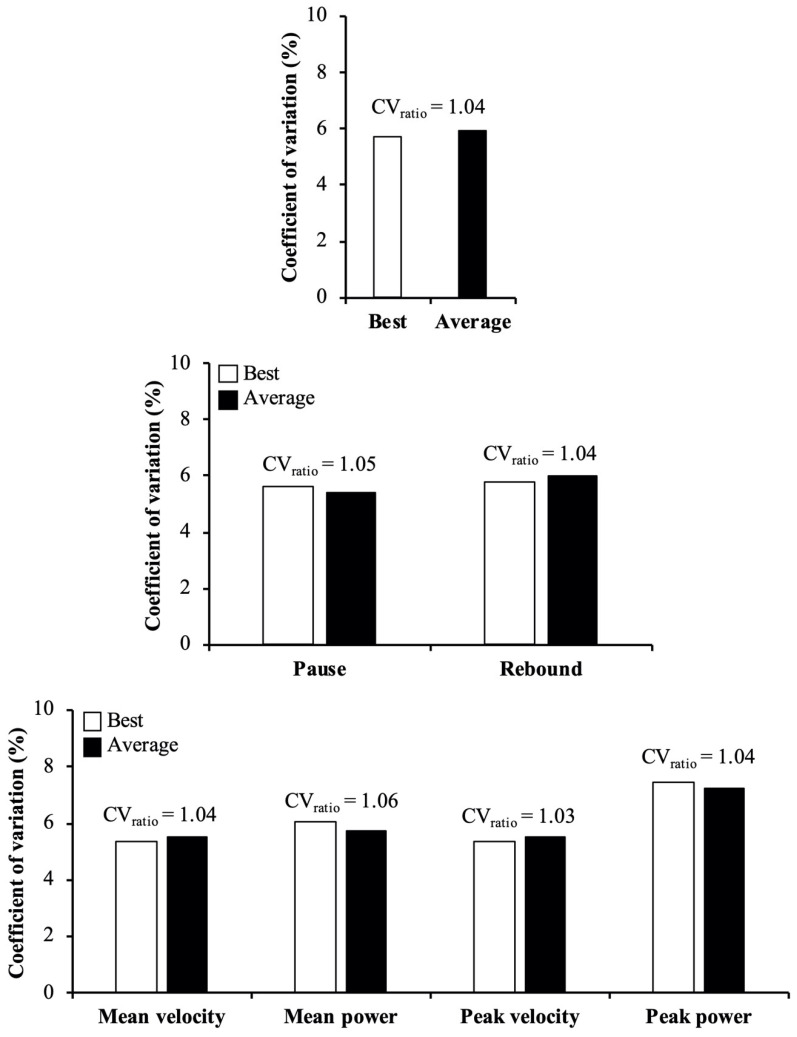
Reliability comparisons between pause (white bars) and rebound (black bars) techniques of the back-squat exercise. Bars represent the median coefficient of variation value obtained combining the 2 repetition criteria and 4 variables (upper panel), the 4 variables separately for each repetition criterion (middle panel), and the 2 repetition criteria separately for each variable (lower panel). Numbers depict the ratio between 2 coefficients of variation (CV_ratio_ = higher/lower value), while meaningful differences in reliability are indicated in bold (CV_ratio_ > 1.15).

**Figure 3 ijerph-18-04626-f003:**
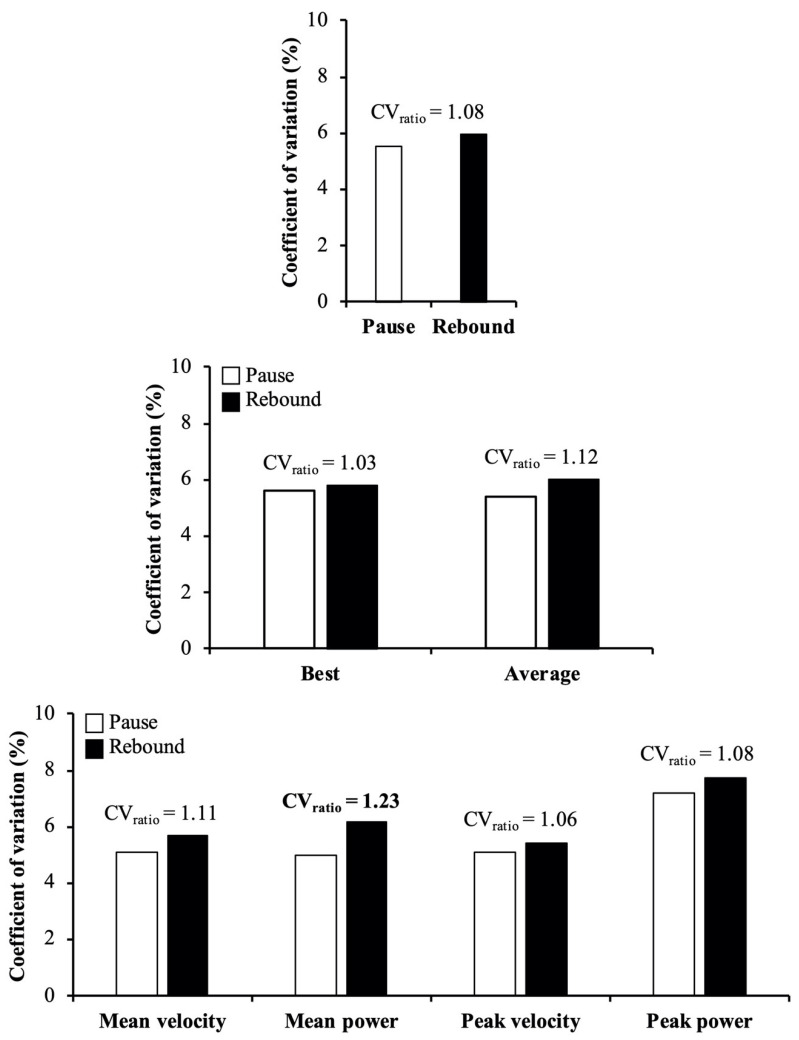
Reliability comparisons between best (white bars) and average (black bars) scores reported during the back-squat exercise. Bars represent the median coefficient of variation value obtained combining the 2 exercise techniques and 4 variables (upper panel), the 4 variables separately for each exercise technique (middle panel), and the 2 exercise techniques separately for each variable (lower panel). Numbers depict the ratio between 2 coefficients of variation (CV_ratio_ = higher/lower value), while meaningful differences in reliability are indicated in bold (CV_ratio_ > 1.15).

**Table 1 ijerph-18-04626-t001:** Between-session reliability of mean velocity, mean power, peak velocity, and peak power variables obtained from the best and average scores of 3 repetitions during the back-squat exercise performed with the pause technique at different loads.

Variable	Load (% 1RM)	Session 1 (Mean ± SD)		Session 2 (Mean ± SD)		SEM (95% CI)		CV (%) (95% CI)		ICC (95% CI)	
		Best Score	Average Score	Best Score	Average Score	Best Score	Average Score	Best Score	Average Score	Best Score	Average Score
Mean velocity (m·s^−1^)	45	0.76 ± 0.05	0.72 ± 0.06	0.75 ± 0.04	0.72 ± 0.05	0.03 (0.02, 0.04)	0.04 (0.03, 0.05)	3.99 (3.13, 5.50)	4.90 (3.84, 6.76)	0.59 (0.26, 0.79)	0.58 (0.25, 0.79)
	55	0.68 ± 0.05	0.65 ± 0.05	0.68 ± 0.04	0.65 ± 0.05	0.03 (0.02, 0.04)	0.03 (0.02, 0.04)	4.50 (3.53, 6.21)	4.81 (3.78, 6.65)	0.62 (0.32, 0.81)	0.62 (0.31, 0.81)
	65	0.60 ± 0.05	0.58 ± 0.05	0.61 ± 0.04	0.58 ± 0.04	0.03 (0.02, 0.04)	0.03 (0.02, 0.04)	4.28 (3.36, 5.91)	5.32 (4.17, 7.34)	0.69 (0.42, 0.85)	0.59 (0.27, 0.79)
	75	0.54 ± 0.06	0.52 ± 0.06	0.56 ± 0.05	0.53 ± 0.05	0.03 (0.02, 0.04)	0.03 (0.02, 0.04)	5.65 (4.43, 7.80)	5.43 (4.26, 7.49)	0.69 (0.42, 0.85)	0.73 (0.48, 0.87)
	85	0.46 ± 0.05	0.44 ± 0.05	0.46 ± 0.05	0.43 ± 0.05	0.03 (0.02, 0.04)	0.03 (0.02, 0.04)	6.05 (4.74, 8.35)	6.93 (5.44, 9.57)	0.68 (0.40, 0.84)	0.67 (0.38, 0.84)
Mean power (W)	45	454 ± 93	435 ± 95	443 ± 88	427 ± 90	20.2 (15.8, 27.8)	21.8 (17.1, 30.1)	4.49 (3.52, 6.20)	5.06 (3.97, 6.99)	0.95 (0.90, 0.98)	0.95 (0.89, 0.98)
	55	505 ± 114	481 ± 105	494 ± 108	475 ± 105	33.2 (26.1, 45.9)	21.6 (17.0, 29.9)	6.65 (5.22, 9.18)	4.53 (3.55, 6.25)	0.92 (0.82, 0.96)	0.96 (0.91, 0.98)
	65	519 ± 115	502 ± 116	527 ± 108	502 ± 104	23.0 (18.0, 31.7)	24.9 (19.5, 34.3)	4.39 (3.45, 6.07)	4.96 (3.89, 6.84)	0.96 (0.91, 0.98)	0.95 (0.90, 0.98)
	75	539 ± 124	514 ± 121	547 ± 118	519 ± 107	29.2 (22.9, 40.4)	24.6 (19.3, 33.9)	5.39 (4.22, 7.44)	4.76 (3.73, 6.57)	0.95 (0.88, 0.98)	0.96 (0.91, 0.98)
	85	518 ± 105	487 ± 105	515 ± 114	483 ± 105	32.5 (25.5, 44.8)	33.3 (26.1, 46.0)	6.28 (4.93, 8.67)	6.87 (5.39, 9.48)	0.92 (0.83, 0.96)	0.91 (0.80, 0.96)
Peak velocity (m·s^−1^)	45	1.34 ± 0.11	1.27 ± 0.12	1.33 ± 0.10	1.28 ± 0.10	0.06 (0.05, 0.09)	0.06 (0.05, 0.08)	4.73 (3.71, 6.53)	4.77 (3.74, 6.59)	0.64 (0.35, 0.82)	0.72 (0.46, 0.86)
	55	1.24 ± 0.10	1.19 ± 0.09	1.23 ± 0.10	1.19 ± 0.09	0.05 (0.04, 0.07)	0.05 (0.04, 0.07)	3.89 (3.05, 5.37)	3.99 (3.13, 5.50)	0.77 (0.56, 0.89)	0.77 (0.54, 0.89)
	65	1.14 ± 0.10	1.11 ± 0.11	1.18 ± 0.08	1.13 ± 0.08	0.06 (0.04, 0.08)	0.06 (0.05, 0.08)	4.93 (3.87, 6.80)	5.29 (4.14, 7.30)	0.60 (0.28, 0.80)	0.62 (0.31, 0.81)
	75	1.07 ± 0.11	1.03 ± 0.11	1.11 ± 0.08	1.06 ± 0.09	0.06 (0.05, 0.08)	0.06 (0.05, 0.08)	5.60 (4.39, 7.73)	5.68 (4.45, 7.83)	0.65 (0.35, 0.82)	0.65 (0.35, 0.82)
	85	0.98 ± 0.08	0.94 ± 0.10	1.01 ± 0.10	0.97 ± 0.10	0.06 (0.05, 0.09)	0.07 (0.05, 0.10)	6.48 (5.08, 8.95)	7.22 (5.66, 9.97)	0.52 (0.18, 0.75)	0.54 (0.20, 0.76)
Peak power (W)	45	984 ± 226	917 ± 229	959 ± 213	910 ± 210	62.8 (49.2, 86.6)	59.2 (46.5, 81.8)	6.46 (5.07, 8.92)	6.49 (5.09, 8.95)	0.92 (0.84, 0.97)	0.93 (0.86, 0.97)
	55	1097 ± 273	1024 ± 251	1060 ± 247	1010 ± 240	79.5 (62.4, 109.8)	58.2 (45.6, 80.3)	7.37 (5.78, 10.18)	5.72 (4.49, 7.90)	0.91 (0.82, 0.96)	0.95 (0.89, 0.98)
	65	1166 ± 271	1110 ± 271	1195 ± 251	1133 ± 248	87.1 (68.3, 120.2)	75.2 (59.0, 103.8)	7.38 (5.78, 10.18)	6.71 (5.26, 9.26)	0.90 (0.78, 0.95)	0.92 (0.84, 0.96)
	75	1245 ± 309	1183 ± 301	1267 ± 282	1206 ± 261	94.9 (74.4, 131.0)	83.3 (65.4, 115.0)	7.55 (5.92, 10.43)	6.98 (5.47, 9.63)	0.90 (0.80, 0.96)	0.92 (0.83, 0.96)
	85	1266 ± 290	1197 ± 285	1311 ± 304	1237 ± 275	110.0 (86.2, 151.8)	105.2 (82.5, 145.3)	8.53 (6.69, 11.78)	8.65 (6.78, 11.94)	0.87 (0.74, 0.94)	0.87 (0.73, 0.94)

1RM, 1-repetition maximum; SD, standard deviation; SEM, standard error of measurement; CV, coefficient of variation; ICC, intraclass correlation coefficient; 95% CI, 95% confidence intervals. Bold numbers indicate an unacceptable reliability (CV > 10% or ICC < 0.70).

**Table 2 ijerph-18-04626-t002:** Between-session reliability of mean velocity, mean power, peak velocity, and peak power variables obtained from the best and average scores of 3 repetitions in the back-squat exercise performed with the rebound technique at different loads.

Variable	Load (% 1RM)	Session 1 (Mean ± SD)		Session 2 (Mean ± SD)		SEM (95% CI)		CV (%) (95% CI)		ICC (95% CI)	
		Best Score	Average Score	Best Score	Average Score	Best Score	Average Score	Best Score	Average Score	Best Score	Average Score
Mean velocity (m·s^−1^)	45	0.78 ± 0.07	0.74 ± 0.07	0.78 ± 0.06	0.75 ± 0.06	0.04 (0.03, 0.06)	0.04 (0.03, 0.06)	5.31 (4.16, 7.32)	5.95 (4.66, 8.21)	0.62 (0.31, 0.81)	0.59 (0.26, 0.79)
	55	0.71 ± 0.07	0.69 ± 0.07	0.72 ± 0.05	0.70 ± 0.05	0.04 (0.03, 0.05)	0.04 (0.03, 0.05)	5.11 (4.01, 7.05)	5.24 (4.11, 7.23)	0.63 (0.33, 0.81)	0.64 (0.34, 0.82)
	65	0.63 ± 0.07	0.61 ± 0.07	0.66 ± 0.06	0.63 ± 0.06	0.03 (0.03, 0.05)	0.03 (0.03, 0.05)	5.35 (4.20, 7.38)	5.60 (4.39, 7.73)	0.73 (0.49, 0.87)	0.69 (0.43, 0.85)
	75	0.56 ± 0.06	0.53 ± 0.06	0.57 ± 0.06	0.55 ± 0.06	0.03 (0.03, 0.04)	0.03 (0.03, 0.04)	5.74 (4.50, 7.92)	5.89 (4.62, 8.14)	0.74 (0.50, 0.88)	0.74 (0.51, 0.88)
	85	0.46 ± 0.07	0.43 ± 0.06	0.48 ± 0.05	0.45 ± 0.05	0.04 (0.03, 0.05)	0.04 (0.03, 0.05)	8.28 (6.49, 11.42)	8.63 (6.77, 11.91)	0.61 (0.29, 0.80)	0.60 (0.29, 0.80)
Mean power (W)	45	467 ± 106	446 ± 99	468 ± 98	450 ± 98±	29.1 (22.8, 40.2)	27.4 (21.5, 37.8)	6.23 (4.88, 8.60)	6.11 (4.79, 8.44)	0.92 (0.84, 0.97)	0.93 (0.85, 0.97)
	55	519 ± 116	500 ± 112	525 ± 106	507 ± 103	32.4 (25.4, 44.7)	31.5 (24.7, 43.5)	6.20 (4.86, 8.56)	6.26 (4.91, 8.64)	0.92 (0.83, 0.96)	0.92 (0.83, 0.96)
	65	545 ± 131	527 ± 128	566 ± 130	547 ± 124	29.8 (23.3, 41.1)	29.3 (23.0, 40.5)	5.35 (4.20, 7.39)	5.46 (4.28, 7.54)	0.95 (0.90, 0.98)	0.95 (0.89, 0.98)
	75	556 ± 133	532 ± 130	564 ± 131	544 ± 128	32.7 (25.7, 45.2)	32.0 (25.1, 44.2)	5.85 (4.59, 8.07)	5.95 (4.67, 8.22)	0.94 (0.88, 0.97)	0.94 (0.88, 0.97)
	85	517 ± 137	485 ± 131	542 ± 122	511 ± 114	48.2 (37.8, 66.5)	46.4 (36.4, 64.0)	9.10 (7.14, 12.57)	9.31 (7.31, 12.86)	0.87 (0.73, 0.94)	0.87 (0.73, 0.94)
Peak velocity (m·s^−1^)	45	1.31 ± 0.14	1.26 ± 0.14	1.32 ± 0.11	1.25 ± 0.10	0.07 (0.06, 0.10)	0.07 (0.06, 0.10)	5.44 (4.26, 7.50)	5.96 (4.67, 8.23)	0.68 (0.40, 0.84)	0.63 (0.33, 0.82)
	55	1.22 ± 0.12	1.17 ± 0.12	1.24 ± 0.09	1.20 ± 0.09	0.07 (0.05, 0.09)	0.06 (0.05, 0.08)	5.40 (4.24, 7.46)	5.02 (3.94, 6.93)	0.65 (0.36, 0.83)	0.72 (0.46, 0.86))
	65	1.14 ± 0.10	1.10 ± 0.11	1.18 ± 0.09	1.14 ± 0.09	0.05 (0.04, 0.08)	0.06 (0.04, 0.08)	4.70 (3.68, 6.48)	5.12 (4.02, 7.07)	0.69 (0.41, 0.85)	0.69 (0.42, 0.85)
	75	1.06 ± 0.11	1.02 ± 0.11	1.10 ± 0.10	1.05 ± 0.11	0.06 (0.04, 0.08)	0.06 (0.05, 0.09)	5.30 (4.16, 7.31)	5.96 (4.67, 8.22)	0.73 (0.48, 0.87)	0.69 (0.43, 0.85)
	85	0.95 ± 0.12	0.90 ± 0.14	1.00 ± 0.09	0.95 ± 0.11	0.06 (0.05, 0.09)	0.08 (0.06, 0.11)	6.44 (5.05, 8.90)	8.50 (6.66, 11.73)	0.69 (0.42, 0.85)	0.60 (0.29, 0.80)
Peak power (W)	45	951 ± 252	893 ± 235	955 ± 212	891 ± 207	76.4 (59.9, 105.4)	76.0 (59.6, 105.0)	8.01 (6.28, 11.06)	8.52 (6.69, 11.77)	0.90 (0.79, 0.95)	0.89 (0.77, 0.95)
	55	1043 ± 259	987 ± 249	1062 ± 230	1010 ± 219	84.2 (66.0, 116.2)	75.1 (58.9, 103.7)	8.00 (6.27, 11.04)	7.53 (5.90, 10.39)	0.89 (0.77, 0.95)	0.90 (0.80, 0.96)
	65	1142 ± 265	1081 ± 263	1185 ± 245	1133 ± 241	55.4 (43.4, 76.5)	67.0 (52.6, 92.5)	4.76 (3.73, 6.57)	6.05 (4.75, 8.35)	0.96 (0.91, 0.98)	0.93 (0.86, 0.97)
	75	1214 ± 297	1149 ± 277	1256 ± 262	1187 ± 251	80.6 (63.2, 111.3)	86.8 (69.1, 119.8)	6.53 (5.12, 9.01)	7.43 (5.82, 10.25)	0.92 (0.84, 0.96)	0.90 (0.79, 0.95)
	85	1197 ± 303	1137 ± 312	1270 ± 248	1192 ± 249	98.2 (77.0, 135.5)	119.8 (93.9, 165.3)	7.96 (6.24, 10.99)	10.29 (8.07, 14.20)	0.88 (0.76, 0.95)	0.83 (0.66, 0.92)

1RM, 1-repetition maximum; SD, standard deviation; SEM, standard error of measurement; CV, coefficient of variation; ICC, intraclass correlation coefficient; 95% CI, 95% confidence intervals. Bold numbers indicate an unacceptable reliability (CV > 10% or ICC < 0.70).

## Data Availability

Raw data of this article are available upon request to corresponding authors.

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
