# Peer review of "Assessment of Back-Squat Performance at Submaximal Loads: Is the Reliability Affected by the Variable, Exercise Technique, or Repetition Criterion?"

_ijerph, 2021, doi:10.3390/ijerph18094626_

Round 1
Reviewer 1 Report
The study is well designed for reliability evaluation of several strength and neuromuscular tests using two different back-squat techniques. The study design includes several various loads in regards percent of 1RM. The outcome is useful for the trainers and athletes to know when evaluation the specific strength progress with a linear encoder. Overall the study is thorough but could have included a physiological rationale for the study results.
Abstract
”.. different mechanical variables..” Mechanical variables seem strange at it is a combination of both mechanics and human biology. The CV includes variations of both mechanic and physiological variables.
L137-138: 3 minutes seems reasonable to resynthesize of ATP-PCr. However PCr is not fully restored and could possibly influence the outcome with a repeated set-up. Please see Sahlin et al, Hultman et al.
L150. How was this controlled that the subjects positioned achieved 90 degree flexion?
Statistical analyses were performed using MBD/MBI. I am sure the authors are aware if the criticism made by other statisticians (see references). Paired t-tests were employed but usually the set-up would be 2x5 repeated measure ANOVA (2 sessions x 5 different %1RM). However, the Cohens d is missing throughout the tables. Please include these numbers. It would be informative to include standard error of measurement’s (part of the CV equation) numbers themselves.
Sainani, K. L. "The Problem with "Magnitude-Based Inference"." Med Sci Sports Exerc 50, no. 10 (Oct 2018): 2166-76.
Curran-Everett, D. "Magnitude-Based Inference: Good Idea but Flawed Approach." Med Sci Sports Exerc 50, no. 10 (Oct 2018): 2164-65.
Borg, D. N., G. M. Minett, I. B. Stewart, and C. C. Drovandi. "Bayesian Methods Might Solve the Problems with Magnitude-Based Inference." Med Sci Sports Exerc 50, no. 12 (Dec 2018): 2609-10.
Welsh, A. H., and E. J. Knight. ""Magnitude-Based Inference": A Statistical Review." Med Sci Sports Exerc 47, no. 4 (Apr 2015): 874-84.
Barker, R. J., and M. R. Schofield. "Inference About Magnitudes of Effects." Int J Sports Physiol Perform 3, no. 4 (Dec 2008): 547-57.
L184: Include the ICC equation.
L215-216: Is it only mechanical?
Table 1 and 2 miss the Cohens d numbers. Further, the outcome of the t-test is missing. Please include these.
Overall the discussion is thorough but has left out the physiological rationale why for example PP is less reliable.
Author Response
COMMENT
The study is well designed for reliability evaluation of several strength and neuromuscular tests using two different back-squat techniques. The study design includes several various loads in regards percent of 1RM. The outcome is useful for the trainers and athletes to know when evaluation the specific strength progress with a linear encoder. Overall the study is thorough but could have included a physiological rationale for the study results.
RESPONSE
The reviewer’s comments are highly appreciated. We have considered all suggestions and we believe our manuscript is stronger as a result of the changes that we have introduced in the revised version of the manuscript.
COMMENT
Abstract. ”.. different mechanical variables..” Mechanical variables seem strange at it is a combination of both mechanics and human biology. The CV includes variations of both mechanic and physiological variables.
RESPONSE
We agree with the reviewer's suggestion. We have replaced "mechanical variables" with "performance variables" throughout the manuscript.
COMMENT
L137-138: 3 minutes seems reasonable to resynthesize of ATP-PCr. However PCr is not fully restored and could possibly influence the outcome with a repeated set-up. Please see Sahlin et al, Hultman et al.
RESPONSE
We understand the reviewer's concern about the rest period. However, 3 and 5 minutes has been recently shown to be enough rest time to maintain high velocities during training sessions not leading to failure (3 sets of five repetitions performed with the back-squat and bench press exercises against 10RM load) (González-Hernández et al., 2020). In this regard, it should be taken into account that a 3-minute rest period would provide a more time-efficient testing protocol of back-squat performance than a 5-minut rest period.
González-Hernández, J.M.; Jimenez-Reyes, P.; Janicijevic, D.; Tufano, J.J.; Marquez, G.; Garcia-Ramos, A. Effect of different interset rest intervals on mean velocity during the squat and bench press exercises. Sports Biomech. 2020. In press.
COMMENT
L150. How was this controlled that the subjects positioned achieved 90 degree flexion?
RESPONSE
We have indicated in the manuscript that “the squat depth was individually controlled using an elastic cord positioned under participants’ hips [31].”
COMMENT
Statistical analyses were performed using MBD/MBI. I am sure the authors are aware if the criticism made by other statisticians (see references). Paired t-tests were employed but usually the set-up would be 2x5 repeated measure ANOVA (2 sessions x 5 different %1RM). However, the Cohens d is missing throughout the tables. Please include these numbers. It would be informative to include standard error of measurement’s (part of the CV equation) numbers themselves.
Sainani, K. L. "The Problem with "Magnitude-Based Inference"." Med Sci Sports Exerc 50, no. 10 (Oct 2018): 2166-76.
Curran-Everett, D. "Magnitude-Based Inference: Good Idea but Flawed Approach." Med Sci Sports Exerc 50, no. 10 (Oct 2018): 2164-65.
Borg, D. N., G. M. Minett, I. B. Stewart, and C. C. Drovandi. "Bayesian Methods Might Solve the Problems with Magnitude-Based Inference." Med Sci Sports Exerc 50, no. 12 (Dec 2018): 2609-10.
Welsh, A. H., and E. J. Knight. ""Magnitude-Based Inference": A Statistical Review." Med Sci Sports Exerc 47, no. 4 (Apr 2015): 874-84.
Barker, R. J., and M. R. Schofield. "Inference About Magnitudes of Effects." Int J Sports Physiol Perform 3, no. 4 (Dec 2008): 547-57.
RESPONSE
We're sorry, but we don't understand the reviewer's concern for statistical analysis. We have simply reported the paired samples t-tests and standardized mean difference (Cohen’s d effect size) to examine the systematic bias of the different performance variables (Atkinson and Nevill, 1998). This information has been presented in the text (lines 192-195) due to the great length of the tables. On the other hand, we have provided the standard error of measurement following the reviewer's indications.
Atkinson, G.; Nevill, A.M. Statistical methods for assessing measurement error (reliability) in variables relevant to sports medicine. Sports Med. 1999, 26, 217-238.
COMMENT
L184: Include the ICC equation.
RESPONSE
As a consequence of the complexity of the ICC equation, we have preferred to report the model directly (3.1). Note that we have also indicated the spreadsheet used for the reliability analysis (Hopkins, 2000).
Hopkins, W. Calculations for reliability (Excel spreedsheet). A New View of Statistics. 2000. Available at: http://www.sportsci.org/resource/stats/relycalc.html#excel. Accessed April 06, 2021.
COMMENT
L215-216: Is it only mechanical?
RESPONSE
We have replaced "mechanical variables" with "performance variables".
COMMENT
Table 1 and 2 miss the Cohens d numbers. Further, the outcome of the t-test is missing. Please include these.
RESPONSE
This information has been provided in the text (lines 193-196) due to the great length of the tables (even more after adding the SEM). Please note that this information is simply to report that there was no systematic bias.
COMMENT
Overall the discussion is thorough but has left out the physiological rationale why for example PP is less reliable.
RESPONSE
We understand the reviewer's concern. However, we believe that it would not be appropriate to give a physiological rationing because (1) we are simply measuring the changes in barbell position, and (2) the power is indirectly obtained from the force calculated from the first and second derivate of the change in barbell position with respect to time. For that reason, we have focused on the external manifestation of the mechanical performance of back-squat exercise – which is measured with a linear position transducer – throughout the manuscript.
Reviewer 2 Report
Dear Authors,
thank you for the submission of your manuscript in IJERPH. I have thoroughly read your study. It has been a great pleasure to review this study.
Article deals with an interesting and debated topic, it is well written, and an accurate and valid methodological setup has been used.
I approve the publication of this paper after minor revision.
However, I have some observations on the paper.
General:
I suggest not to use subject when you speak about people, it sounds very impersonal. In this case, I suggest “participant or person”.
Title: ok
Abstract: Well written
Introduction:
L.49 Change “not” with “no”
L.57-59 You passed from squat to back-squat without explaining the difference. I suggest to rephrase these sentences adding back-squat as a loaded variable of the squat.
L.68-70 This sentence seems not to be in contrast with the previous one. I suggest to change “However” with “In addition” or “Moreover”.
Materials and Methods:
L.113-117 You did not include the Ethical committee statement for the study. Please add it.
L.158 The sentence is redundant “(i.e., no pause between the lowering and lifting phases)”. I suggest removing it.
It could be useful to show the set-up for the acquisition. I suggest adding a figure.
Results: ok
Discussion:
L.277-279 What do you mean by “practitioners should be more careful when tracking changes in mechanical performance against heavy loads (≥ 85%1RM) during the back-squat exercise”? What is the practical suggestion you give about this?
Conclusion:
- 368 Why only the best score and not also the worst score?
Author Response
COMMENT
Dear Authors,
thank you for the submission of your manuscript in IJERPH. I have thoroughly read your study. It has been a great pleasure to review this study.
Article deals with an interesting and debated topic, it is well written, and an accurate and valid methodological setup has been used.
I approve the publication of this paper after minor revision.
However, I have some observations on the paper.
RESPONSE
The reviewer’s comments are highly appreciated. We have considered all suggestions and we believe our manuscript is stronger as a result of the changes that we have introduced in the revised version of the manuscript.
COMMENT
General: I suggest not to use subject when you speak about people, it sounds very impersonal. In this case, I suggest “participant or person”.
RESPONSE
We have replaced "subject" with "participant" throughout the manuscript.
COMMENT
Title: ok
RESPONSE
The reviewer’s comment is highly appreciated.
COMMENT
Abstract: Well written
RESPONSE
The reviewer’s comment is highly appreciated.
COMMENT
Introduction: L.49 Change “not” with “no”
RESPONSE
This change has been made.
COMMENT
Introduction: L.57-59 You passed from squat to back-squat without explaining the difference. I suggest to rephrase these sentences adding back-squat as a loaded variable of the squat.
RESPONSE
We have clarified that we are referring to the back-squat in the revised version of the manuscript. The main reason for indicating back-squat rather than just squat is because other variants of the squat can be performed during resistance training (e.g., front squat).
COMMENT
Introduction: L.68-70 This sentence seems not to be in contrast with the previous one. I suggest to change “However” with “In addition” or “Moreover”.
RESPONSE
This change has been made.
COMMENT
Materials and Methods: L.113-117 You did not include the Ethical committee statement for the study. Please add it.
RESPONSE
This information has been added in the revised version of the manuscript.
COMMENT
Materials and Methods: L.158 The sentence is redundant “(i.e., no pause between the lowering and lifting phases)”. I suggest removing it.
It could be useful to show the set-up for the acquisition. I suggest adding a figure.
RESPONSE
This information has been removed from the revised version of the manuscript. On the other hand, we believe that we have provided all the detailed information of the acquisition in the manuscript and, therefore, a figure would not provide any additional information. Note also that we have already provided a reasonable number of tables/figures (2 and 3, respectively) in the manuscript.
COMMENT
Results: ok
RESPONSE
The reviewer’s comment is highly appreciated.
COMMENT
Discussion: L.277-279 What do you mean by “practitioners should be more careful when tracking changes in mechanical performance against heavy loads (≥ 85%1RM) during the back-squat exercise”? What is the practical suggestion you give about this?
RESPONSE
This information has been clarified in the revised version of the manuscript and it now reads: “These results collectively suggest that practitioners should be more careful when tracking changes in mechanical performance against heavy loads (≥ 85%1RM) during the back-squat exercise due to lower reproducibility of measurement”. It should be noted that, in line with previous research (Thompson et al., 2021), the results of the present study show that the absolute reliability was acceptable and consistent under light-moderate loads (45-75%1RM), but it was slightly lower under heavy loads (85%1RM).
Thompson, S.W.; Rogerson, D.; Ruddock, A.; Banyard, H.G.; Barnes, A. Pooled versus individualized load–velocity profiling in the free-weight back squat and power clean. Int J Sports Physiol Perform. 2021. In press.
COMMENT
Conclusion: 368 Why only the best score and not also the worst score.
RESPONSE
The repetition criterion based on the best score is frequently used in the scientific literature to represent the maximum mechanical performance of an individual (Ruf et al., 2018; Dorrell et al., 2019; Hughes et al., 2019; Pérez-Castilla et al., 2020). This is the main reason why the worst score is rarely reported in research.
Ruf, L.; Chery, C.; Taylor, K.L. Validity and reliability of the load-velocity relationship to predict the one-repetition maximum in deadlift. J Strength Cond Res. 2018, 32, 681-689.
Dorrell, H.F.; Moore, J.M.; Smith, M.F.; Gee, T.I. Validity and reliability of a linear positional transducer across commonly practised resistance training exercises. J Sports Sci. 2019, 37, 67-73.
Hughes, L.J.; Banyard, H.G.; Dempsey, A.R.; Scott, B.R. Using a load-velocity relationship to predict one repetition maximum in free-weight exercise. J Strength Cond Res. 2019, 33, 2409-2419.
Pérez-Castilla, A.; Martínez-García, D.; Jerez-Mayorga, D.; Rodríguez-Perea, Á.; Chirosa-Ríos, L.J.; García-Ramos, A. Influence of the grip width on the reliability and magnitude of different velocity variables during the bench press exercise. Eur J Sport Sci. 2020, 20, 1-10.
Round 2
Reviewer 1 Report
The revision of the manuscript is well revised and further justified by the authors in their replies point by point.